# First field estimation of greenhouse gas release from European soil-dwelling Scarabaeidae larvae targeting the genus *Melolontha*

Carolyn-Monika Görres[1,2]*, Claudia Kammann[1]

**1** Department of Applied Ecology, Hochschule Geisenheim University, Geisenheim, Germany, **2** Department of Soil Science and Plant Nutrition, Hochschule Geisenheim University, Geisenheim, Germany

* carolyn.goerres@hs-gm.de

**Data Availability Statement:** All data are within the manuscript and its Supporting Information files.

**Funding:** CMG: This project has received funding from the European Union's Horizon 2020 research

## Abstract

Arthropods are a major soil fauna group, and have the potential to substantially influence the spatial and temporal variability of soil greenhouse gas (GHG) sinks and sources. The overall effect of soil-inhabiting arthropods on soil GHG fluxes still remains poorly quantified since the majority of the available data comes from laboratory experiments, is often controversial, and has been limited to a few species. The main objective of this study was to provide first insights into field-level carbon dioxide ($CO_2$), methane ($CH_4$), and nitrous oxide ($N_2O$) release of soil-inhabiting larvae of the Scarabaeidae family. Larvae of the genus *Melolontha* were excavated at various sites in west-central and southern Germany, covering a wide range of different larval developmental stages, larval activity levels, and vegetation types. Excavated larvae were immediately incubated in the field to measure their GHG production. Gaseous carbon release of individual larvae showed a large inter- and intra-site variability which was strongly correlated to larval biomass. This correlation persisted when upscaling individual $CO_2$ and $CH_4$ production to the plot scale. Field release estimates for *Melolontha* spp. were subsequently upscaled to the European level to derive the first regional GHG release estimates for members of the Scarabaeidae family. Estimates ranged between 10.42 and 409.53 kt $CO_2$ $yr^{-1}$, and 0.01 and 1.36 kt $CH_4$ $yr^{-1}$. Larval $N_2O$ release was only sporadically observed and not upscaled. For one site, a comparison of field- and laboratory-based GHG production measurements was conducted to assess potential biases introduced by transferring Scarabaeidae larvae to artificial environments. Release strength and variability of captive larvae decreased significantly within two weeks and the correlation between larval biomass and gaseous carbon production disappeared, highlighting the importance of field measurements. Overall, our data show that Scarabaeidae larvae can be significant soil GHG sources and should not be neglected in soil GHG flux research.

and innovation programme under the Marie Sklodowska-Curie grant agreement No 703107. The funders had no role in study design, data collection and analysis, decision to publish, or preparation of the manuscript. Funder website: https://ec.europa.eu/research/mariecurieactions/actions/individual-fellowships_en.

**Competing interests:** The authors have declared that no competing interests exist.

## Introduction

A precise knowledge of the sink and source distributions of greenhouse gases (GHG) in regional and global carbon (C) and nitrogen (N) budgets, and of the processes governing them, is a necessary prerequisite for the development and assessment of climate change adaptation and mitigation strategies [1–3]. A major challenge in climate change research is to fully understand and accurately quantify interannual to decadal variabilities in atmospheric GHG concentrations driven by natural processes [1,4]. Soils can act as both important sinks and sources of atmospheric carbon dioxide ($CO_2$), methane ($CH_4$), and nitrous oxide ($N_2O$), but are also components in the global C and N budget with large uncertainty estimates [1–3]. In recent years, it has been proposed to reduce these uncertainties by shifting from an implicit to an explicit representation of soil biota in ecosystem and, ultimately, Earth system models [5–7]. However, a major constraint for developing such new biogeochemical models is the lack of field data, especially for soil biota groups other than microorganisms [6,7], in this study referred to as soil fauna.

Soil fauna can substantially influence the spatial and temporal variability of GHG sinks and sources in the field [6,8–10]. They directly contribute to soil GHG fluxes via their respiratory and metabolic activities and indirectly by changing the physical, chemical and biological properties of soils through bioturbation, fragmentation and redistribution of plant residues, defecation, soil aggregate formation, herbivory, and grazing on microorganisms and fungi [6,11]. Climate, abiotic soil conditions, land management, and interactions within the soil food web modify their abundance, activity, and vertical and horizontal distribution in soils, and thus their contribution to soil GHG fluxes [6,8–10]. However, the magnitude of the effect of soil fauna on the overall GHG sink and source capacity of soils remains poorly quantified since the majority of our current knowledge still comes from laboratory experiments, is often controversial, and has been limited to only a few regions and species [12–15].

A considerable portion of soil-inhabiting animals belongs to the phylum Arthropoda [16]. Certain soil-inhabiting Arthropoda groups–termites, scarab beetles, millipedes, and cockroaches–have received special attention as GHG producers mainly due to their ability to emit $CH_4$. However, apart from termites, their GHG production has never been studied in the field and quantified at different scales [12,13]. Of this group, termites are the only soil-inhabiting Arthropoda, which have been explicitly considered as significant GHG source both on a global and regional level thus far [12,17]. They are assumed to contribute about 1–3% to the global annual $CH_4$ budget, but the large variation in emission estimates in the literature (0.9 to 150 Tg $CH_4$ $y^{-1}$) underlines the uncertainty in the available data sets [2,12,18]. The overall aim of this study was to provide first insights into field-level $CO_2$, $CH_4$, and $N_2O$ release of soil-inhabiting larvae of the Scarabaeidae family. We conducted the first explorative pilot GHG field measurement study on this Arthropoda group, focusing on larvae of the genus *Melolontha* covering a wide range of different larval developmental stages, larval activity levels, and vegetation types in west-central and southern Germany. Field release estimates for *Melolontha* spp. were subsequently upscaled to the European level to derive the first regional GHG estimate for members of the Scarabaeidae family. In addition, a comparison of field- and laboratory-based GHG production measurements was conducted to assess potential biases introduced by transferring Scarabaeidae larvae to an artificial environment for extended time periods.

## Materials and methods

### Sampling sites and species

This study was conducted in central and southern Germany–a temperate climate region with average annual air temperatures between 8 and 12˚C and average annual precipitation

between 600 and 1000 mm (reference period 1961–1990) [19]. Target species were the Common cockchafer (*Melolontha melolontha*) and the Forest cockchafer (*M. hippocastani*) because they have a soil pest status in Europe and they represent two different distribution patterns of Scarabaeidae larvae in soils. Due to the pest status, they are one of the few European Scarabaeidae species for which regular monitoring programs exist, and thus, knowledge on larval ecology is relatively good [20,21]. *M. melolontha* and *M. hippocastani* live three and four years as root-feeding larvae in soils, respectively, progressing through three larval instars before pupating and evolving into adult beetles. Larval size can reach up to 65 mm [22]. The majority of individuals of a local population is of the same age, and population sizes tend to increase with every completed life cycle. *M. melolontha* inhabits open landscapes (e.g. pastures, vegetable crops, orchards, vineyards) and feeds in the rhizosphere mainly at 0–10 cm soil depth, while *M. hippocastani* inhabits forests and has a wider vertical soil profile distribution (0–40 cm soil depth) following tree root distribution [20,23,24]. During the 2017 vegetation period, larvae were excavated at six different sites covering a wide range of larval developmental stages and environmental conditions. Those sites were a meadow with *Beauveria* spp. infestation (Site 1), a Christmas tree plantation (tree height < 1 m) (Site 2), a meadow without *Beauveria* spp. infestation (Site 3), and three mixed deciduous forest sites (Sites 4–6) (see S1 File for more details).

## Soil excavations

At each site, two to four randomly chosen plots with an area of 50 cm × 50 cm were carefully excavated by hand to a depth of ~50 cm depending on site conditions. All larvae encountered in a plot were collected for measurements. For each excavated larva, the following properties were recorded: excavation depth, weight, species, and instar. For species and instar identification, we relied on local expert opinion, but even for experts it was not always possible to distinguish between the two *Melolontha* species in the field. The adult beetles are easily identifiable as either *M. melolontha* or *M. hippocastani*; however, the larvae themselves can only be identified to the genus level based on morphological features alone [25]. For each plot, soil temperature and soil moisture were measured at the soil surface (0–5 cm depth) and at the plot's final excavation depth (HH2 moisture meter with WET sensor, Delta-T Devices Ltd., Cambridge, United Kingdom). Air temperature and air pressure were measured with a handheld weather station (SM-28 Skymate PRO, WeatherHawk, Logan, UT, USA) at ~2 m height at each plot. Total larval abundance $m^{-2}$ for each plot was estimated by multiplying the number of collected larvae by four, since a plot covered an area of 0.25 $m^2$.

## Gas production measurements

In the context of this study, the term "production" is used interchangeably with the terms "release" and "emission". Thus, "emission" is used in its broader definition of simply the direct release of gas from a larva, and not *sensu stricto* as just the gas release into the atmosphere over a specified area and specified time interval.

Immediately following soil excavation, the larvae were individually placed in 110 ml glass tubes, which were sealed air-tight with butyl rubber stoppers. All glass tubes were extensively waved in ambient air prior to the incubations to assure equal starting GHG concentrations across all tubes. The larvae were incubated in the field for about an hour and a blank measurement (i.e. a sealed glass tube without a larva) was included at each plot. During this time, the incubation tubes were shaded and placed onto the soil surface next to each plot. Larvae were not cleaned prior to the incubation since no soil particles adhered to their skin, but larvae could defecate during incubation. At the end of the incubation period, 25 ml of air were

extracted from each glass tube with a plastic syringe and transferred to evacuated 12 ml glass vials sealed with grey chlorobutyl rubber septa (Labco Exetainers 839W, Labco Limited, Lampeter, United Kingdom). In addition, syringe samples of ambient air were collected during field incubations, including incubation starts, and also stored in Exetainers. At Site 3, in addition to 18 larvae incubated directly in the field, 65 larvae were excavated and transferred to the laboratory, instead of being field-incubated. These larvae were kept individually in small plastic containers filled with ~100 ml soil from the excavation site, and were supplied with ample amounts of fresh grass roots as well as carrot slices as food sources. Storage temperature was 18˚C and soils were sprayed with tap water once per week to keep them moist. After 7 days, 39 of these larvae were incubated in the laboratory (incubation temperature 24˚C) following the same protocol as in the field. After another 11 days, the remaining 26 larvae were incubated for gas sample collection. Gas samples were analysed with a SRI 8610C gas chromatograph with autosampler (SRI Instruments Europe GmbH, Bad Honnef, Germany) equipped with a flame ionisation detector (FID) coupled to a methanizer for $CO_2$ and $CH_4$ measurements, and an electron capture detector (ECD) for $N_2O$ measurements. Each detector was equipped with a Porapak Q pre-column (stainless steel tubing, length 1 m, 1/8 in. OD, 2 mm ID) and a Hayesep D column (stainless steel tubing, length 3 m, 1/8 in. OD, 2 mm ID). Column oven and detector temperatures were 70˚C and 330˚C, respectively. Nitrogen 5.0 ($N_2$) was supplied as carrier gas at a pressure of 20 PSI (138 kPa). The ECD was additionally supplied with a make-up gas (4.5% $CO_2$ in $N_2$ 5.0) [26] at a pressure of 3 PSI (21 kPa). Due to temporary ECD failure, no $N_2O$ data are available for site 1 (S1 File). Peak integrations in the chromatograms were performed with PeakSimple Version 4.39 (6 channel) (SRI Instruments). Calibration curves were recorded with a 4-point standard gas series ranging from 304.0 to 3999.6 ppm $CO_2$, 1.0 to 20.9 ppm $CH_4$, and 248.4 to 15100.0 ppb $N_2O$, respectively. Concentration increases during the vial incubation time averaged over all samples were 3037.3±1830.2 ppm for $CO_2$, 22.1 ±20.7 ppm for $CH_4$, and 11.1±51.1 ppb for $N_2O$ (average ± standard deviation). Gas concentrations for each individual sample are listed in the supporting information (S1 File).

## Data processing and analysis

For each site, the $CO_2$, $CH_4$, and $N_2O$ emissions of each individual larva were quantified by subtracting the respective average incubation blank gas concentration value from each larval sample gas concentration value, applying the Ideal Gas Law [27], and normalizing by incubation time. Prior to these calculations, the blank measurements had been compared to the Exetainers containing ambient air samples to make sure that measurements were not influenced by the used rubber stoppers, and that the waving of the glass tubes prior to the incubations had been effective. The relative error for each gas emission value was estimated via error propagation assuming the following random errors: 2 Kelvin for air temperature, 10 kPa for air pressure, and 0.01 L for the incubation volume. The random error for the blank-corrected larval sample gas concentration values ($CO_2$ and $CH_4$ in ppm, $N_2O$ in ppb) was the propagated error of the uncorrected larval sample gas concentration value and the incubation blank gas concentration value uncertainties derived from the gas chromatographic calibration curves. The complete flux calculation procedure starting from the raw data is included in S1 File. The relative propagated error for the larval $CO_2$ and $CH_4$ emission estimates ranged between 13 and 16%, apart from a few exceptions. The relative propagated error for the final larval $N_2O$ emission estimates varied widely as the majority of $N_2O$ emissions was not significant. Emissions were classified as non-significant when the propagated error estimate exceeded the emission estimate.

A test for correlations between paired samples was performed on the entire pooled larval field dataset using Spearman's *rho* ($r_s$) statistic. Input variables were instar, larval excavation depth, larval weight (= biomass), larval abundance at the respective plot, individual $CO_2$, $CH_4$, and $N_2O$ emissions, air temperature (= incubation temperature), plot soil surface temperature and moisture, soil temperature and moisture at the respective plot bottom, time of day during which the incubation took place, and incubation duration. For the comparison of larval field- and laboratory-measured emissions from Site 3, the same test statistic was also applied to the three flux data subsets (0, 7 and 18 days after excavation). The test was performed separately on each data subset with larval weight and larval $CO_2$, $CH_4$, and $N_2O$ emissions as input variables. Across-group comparisons on the data subsets were only carried out on larval weight to check for significant differences in mean biomass (Kruskal-Wallis rank sum test).

Larval $CO_2$ and $CH_4$ emissions were scaled up in two steps: from individual larvae to the plot level and from plot level to European level. Total larval emissions and larval biomass per plot were calculated by summing up the individual emissions or biomass and multiplying by four to scale to 1 $m^2$. Larval biomass was subsequently used as independent variable in linear regression analysis for modelling $m^2$-level larval $CO_2$ and $CH_4$ emissions. Linear mixed models were considered for the analysis of the emission data both on the individual and plot level, but yielded no further insights. The obtained plot level larval $CO_2$ and $CH_4$ emissions estimates were upscaled for 6 months per year (excluding larval winter rest) with the available literature data on European land area colonised by *Melolontha* spp. (200,000 ha [20]) to derive a first rough annual $CO_2$ and $CH_4$ emission range estimate for Europe. This was a very simplistic upscaling approach. Employing a more advanced and precise upscaling approach was not yet possible, due to lack of field data on both larval biomass and larval emissions, as well as the unbalanced design of our field dataset. In addition, it needs to be considered that the available literature values on colonised land area are likely very conservative [20]. Due to their scattered occurrence, $N_2O$ emissions were not upscaled.

All test statistics and regression analysis were performed with the software R (version 3.4.3) [28]. In addition to the software's standard library, the function 'chart.Correlation' (package: PerformanceAnalytics) [29] was used.

## Results

Gaseous carbon emissions of individual *Melolontha* spp. larvae showed a large inter- and intra-site variability which could not be explained by differences in soil temperature (range: 11.4–29.3˚C) and soil moisture (range: 3.2–32.7 vol%), or incubation duration. Correlations between emissions and time of day of the incubation and incubation temperature, respectively, were weak ($r_s$ < 0.35) (S1 Fig). There was a clear tendency for emissions to increase with larval biomass at the site level, especially for $CO_2$ (Fig 1). Average larval biomass ranged between 0.5 and 2.2 g larva$^{-1}$. When pooling all data regardless of site and species, there was a strong positive correlation between $CO_2$ and $CH_4$ emissions ($r_s$ = 0.76, p<0.001), and between larval biomass and $CO_2$ emissions ($r_s$ = 0.84, p<0.001). The correlation between larval biomass and $CH_4$ emissions was less pronounced, but still significant ($r_s$ = 0.68, p<0.001). The excavation depth of the larvae correlated negatively with larval biomass ($r_s$ = -0.51, p<0.001), $CO_2$ emissions ($r_s$ = -0.58, p<0.001), and $CH_4$ emissions ($r_s$ = -0.48, p<0.001), respectively (S1 Fig). It could be seen as an indicator for larval access to fresh plant root material, which was higher the closer the larvae were to the soil surface, i.e. the lower the excavation depth was. Two-thirds of the larvae were found at 0–15 cm soil depth. Nitrous oxide emissions were only occasionally observed. Out of the 64 field larvae tested for $N_2O$ (sites 2–6), 13 individuals emitted significant amounts, ranging between 1.3 and 90.4 ng $N_2O$ h$^{-1}$ larva$^{-1}$.

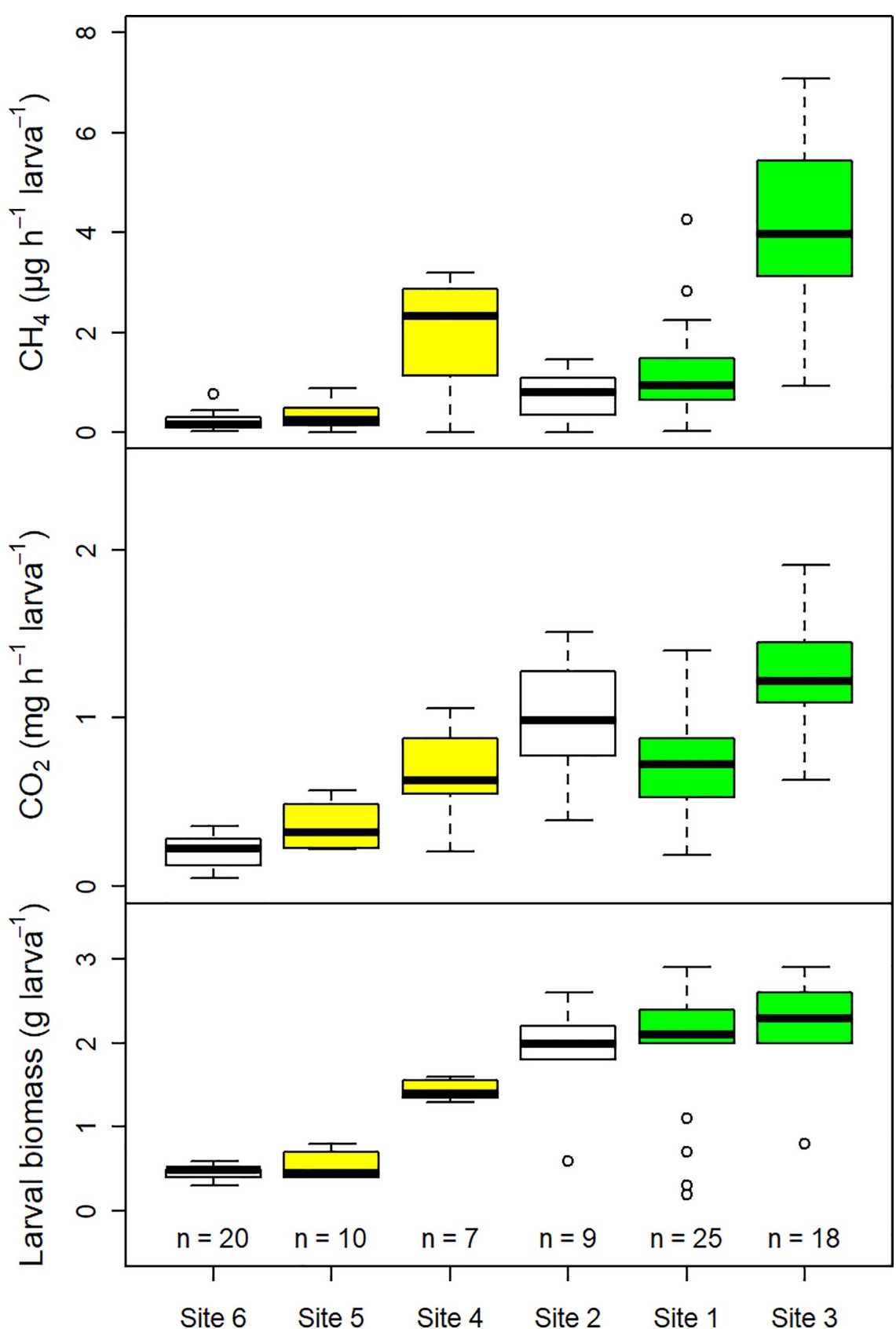

**Fig 1. Direct CH₄ and CO₂ emissions, and larval biomass of individual *Melolontha* spp. larvae during field incubations.** The box midline shows the median, with the upper and lower limits of the box being the 75th and 25th percentile, respectively. The whiskers extend up to 1.5 times from the box edges to the furthest data point within that distance. Sampling sites are sorted by average larval biomass in ascending order. The colours of the boxplots indicate the species (white = *Melolontha* spp., green = *M. melolontha*, yellow = *M. hippocastani*).

Since Scarabaeidae larvae need to reach a certain biomass to be able to pupate and since their food intake increases with size, larval biomass was a good proxy for larval age, sampling time, and food availability. Larval biomass could also be used to differentiate between species and to encode larval abundances at the plot scale. Larval abundances ranged between 4 and 68 larvae m$^{-2}$ (S1 File). The correlation between gaseous carbon emissions release and larval biomass persisted when upscaling emissions to the plot scale, and a large proportion of the inter- and intra-site emission variability could be explained by variations in total larval biomass (Fig 2). Across all sites, $CO_2$ emissions increased on average by 0.51±0.03 mg $CO_2$ h$^{-1}$ m$^{-2}$ with each g total larval biomass increase (p<0.001). The relationship between CH$_4$ emissions and total larval biomass was best fitted with a linear regression model including a polynomial term (S2 File). Plot-scale emissions ranged between 1.19 and 46.75 mg $CO_2$ h$^{-1}$ m$^{-2}$, and 1.15 and 155.58 µg CH$_4$ m$^{-2}$ h$^{-1}$ (excluding one plot at Site 4 with zero emissions) (Fig 2). Based on these values and a literature value of 200,000 ha colonised by *Melolontha* spp. in Europe, total gaseous carbon emissions from European *Melolontha* spp. larvae alone were estimated to range between 10.42 and 409.53 kt $CO_2$ yr$^{-1}$, and 0.01 and 1.36 kt CH$_4$ yr$^{-1}$.

A comparison of field and laboratory measurements from larvae excavated at Site 3 revealed a strong impact of laboratory conditions on $CO_2$ and CH$_4$ emissions. Despite no significant differences in larval biomass between the three measured groups (p = 0.12) and ample food supply, overall emission strength and variability decreased rapidly with prolonged time at the laboratory (Fig 3). In contrast to the field observations, no significant correlation between larval biomass and $CO_2$ and CH$_4$ emissions was found, respectively, after two and a half weeks in

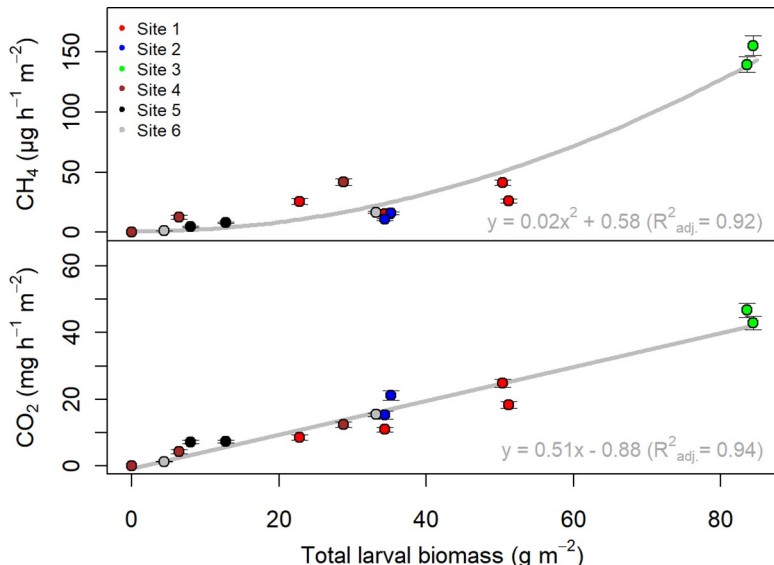

**Fig 2. Cumulated larval CO₂ and CH₄ emission estimates from field measurements for each individual sampling plot in relation to total cumulated larval biomass grouped by sampling site.** Error bars represent the propagated error of the individual larval emissions. Results of linear regression analysis using the complete dataset are given in the respective subfigure for CH$_4$ and $CO_2$.

the laboratory. $N_2O$ emissions tended to be lower under laboratory conditions in comparison to field measurements as well; however, it was not possible to discern a statistically significant effect of the laboratory conditions on larval $N_2O$ emissions. Of the 65 larvae incubated in the laboratory, only 13 emitted $N_2O$, with emissions ranging between 1.81 and 43.70 ng $N_2O$ h$^{-1}$ larva$^{-1}$ (S1 File).

## Discussion

There are no field studies on direct $CO_2$ and $CH_4$ emissions from soil-inhabiting Scarabaeidae larvae yet to which we can compare our data [12], but both the field- and laboratory-measured emission rates fall within the range of emission rates known from the few available laboratory studies on Scarabaeidae larvae and other soil Arthropoda groups in temperate regions [13,30,31], or field and laboratory studies on termites in temperate, subtropical and tropical regions [12,32,33]. However, our study demonstrates how careful we have to be in interpreting GHG emission rates derived from laboratory studies. It has been known for termites that emission rates can decline over the course of a laboratory experiment [34]. In addition, we show that such a trend can also coincide with a considerable reduction of the emission rate variability between individual larvae and a disappearance of correlations between emission rates and environmental variables in comparison to field measurements. Significant changes in larval metabolism due to the changes in environments–which affected absolute soil temperature and moisture values as well as their diurnal variations, oxygen supply, soil porosity, and food supply–are potential underlying causes for these observations. The transport from the field to the laboratory presented an additional source of stress. Such changes can not only affect the production rate of GHG within larvae, but also GHG release rates [35,36]. Larvae can release gases via two pathways: the respiratory system (i.e. tracheal tubes and spiracles) and the anus. For $CO_2$ it is certainly a mixture of both pathways, whereas gases produced in the digestive tract like $CH_4$ and $N_2O$ might be primarily released via the anus [37,38]. At which ratios these pathways occur under varying environmental conditions for different GHG is still unknown [35,38]. For the anus pathway, it also has to be considered if physically handling the animals leads to significant stress-induced degassing of the hindguts before larvae are sealed airtight inside incubation vessels. Changes in environmental conditions also occur to varying degrees and at different temporal resolutions during field incubations, and thus, there is certainly a measurement bias in the field GHG emissions estimates as well. However, we do not know the magnitude of this bias and in how far (potentially) co-occurring processes affecting both larval GHG production and release rates might cancel or amplify each other [35,37].

   Large variations in emission rates and the use of larval biomass for emission rate upscaling are features well known from termite studies [33,34,39,40]. Our biomass-based European $CO_2$ and $CH_4$ emission estimates for *Melolontha* spp. larvae are two orders of magnitude lower than the corresponding emission estimates for termites in temperate regions [39]. Termites are considered as a significant, but quite small global GHG source with the majority of these emissions stemming from subtropical and tropical regions [12,17,41]. Greenhouse gas emissions of soil Arthropoda groups other than termites are seen as too low to significantly affect regional budgets [12,13], which our study seems to confirm at the first glance. However, in contrast to many termite studies, we did not attempt to use the emission rates of a few species to infer the emissions for this entire Arthropoda group. Our estimates are strictly for the genus *Melolontha* only, and thus, show only a fraction of the GHG emission potential of European Scarabaeidae larvae. Furthermore, the available data estimates on European land area colonised by *Melolontha* spp. are incomplete since they are based on surveys from only a few

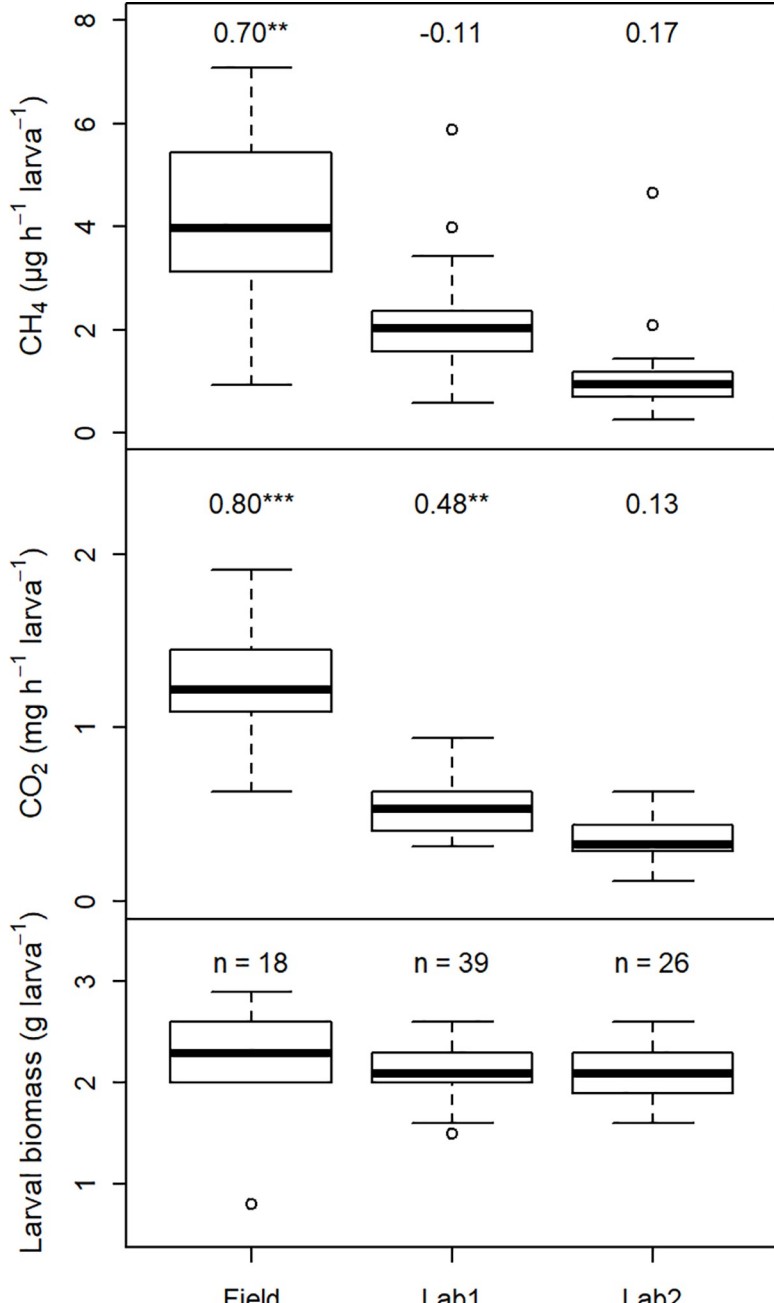

**Fig 3. Comparison of direct *M. melolontha* larval $CO_2$ and $CH_4$ emissions from field and laboratory measurements.** 83 larvae were excavated at sampling Site 3 on 26.05.2017. Larval emissions of batch "Field" were sampled in the field directly after excavation. Batch "Lab1" and "Lab2" were kept 7 and 18 days in the laboratory, respectively, before emissions were measured. The boxplots for "Field" are identical to the boxplots for Site 3 in Fig 1. Numbers above the $CO_2$ and $CH_4$ emissions box plots are Spearman correlation coefficients between the respective gas emissions and the batch's larval weight.

European countries (Austria, Belgium, Czech Republic, Denmark, France, Germany, Italy, Poland, and Switzerland) [20].

Worldwide, larvae of several Scarabaeidae species are regarded as economically important pest insects [42,43]. Regionally, these pest insects can reach biomass levels comparable to or

considerably higher than those used for upscaling termite GHG emissions [21], but we have no estimates of the total biomass of soil-dwelling Scarabaeidae larvae in Europe. Furthermore, for $CH_4$ it is important to differentiate between gross and net soil fluxes. Purely biomass-based $CH_4$ emission rates like ours represent gross soil $CH_4$ emissions, as they do not account for simultaneously occurring gross $CH_4$ consumption rates in soils [18,39,41]. Recent studies suggest that $CH_4$ emissions of soil-inhabiting Scarabaeidae larvae can stimulate soil $CH_4$ consumption, and thus, potentially increase the overall net $CH_4$ sink capacity of upland soils [44,45]. These larvae can be considered as $CH_4$-emitting hot spots in soils, and the magnitude of the stimulation of $CH_4$ consumption around these hot spots will depend on their abundance, longevity and $CH_4$ emission strength, as well as on environmental variables influencing soil gas diffusivity (e.g. soil temperature and moisture, soil porosity and bulk density) [46–48]. For regional and global $CH_4$ budgets, it might therefore be more important to quantify the effect of soil faunal $CH_4$ emissions on the net soil $CH_4$ flux, instead of just quantifying total soil faunal $CH_4$ emissions [18]. However, it is to date unknown if and to what extent stimulated $CH_4$ consumption may or may not act as a biofilter for larval $CH_4$-emitting hot spots, i.e. if the outcome may be an increased net $CH_4$ efflux, or sometimes even an increased net $CH_4$ consumption in Scarabaeidae larvae-infested soils; and how soil properties can shape the net outcome. Regarding the larval potential to turn sites into (temporary) net soil $CH_4$ sources, *Melolontha melolontha* larvae feed on roots directly below the turf, i.e. less than 5 cm below the soil surface. Under these circumstances, the gas diffusion path through the soil might be too short to result in significant mitigations of larval $CH_4$ emissions through $CH_4$ oxidizers. For gaseous carbon emissions of *Melolontha hippocastani* larvae an important question to consider is if they can be channelled through tree roots and stems directly to the atmosphere [49]. To be able to answer all these questions, it is imperative to further the development of non-invasive *in-situ* tools for studying gross $CH_4$ fluxes in soils, e.g. stable carbon isotopes measurement techniques [18,50–52]. These tools are also essential to assess any potential biases introduced by the current standard incubation method, also used in this study, for quantifying larval GHG emissions.

Regarding $N_2O$, the emission rates measured in this study were of the same magnitude as those observed from earthworms [53]. There are also a few studies available from other soil faunal groups, e.g. ants [54] and one laboratory-based study on Scarabaeidae larvae [55], but earthworms are the only faunal group for which a considerable amount of literature on soil $N_2O$ emissions is available [9]. Earthworms are not host to an endemic denitrifier community in their gut system, but in their presence $N_2O$ emissions can increase by more than 40% due to the activation of ingested nitrate- and nitrite-reducing soil bacteria during earthworm gut passage [14]. It is unclear if Scarabaeidae larvae are capable to affect soil $N_2O$ emissions in a similar manner, but the study by Majeed et al. [55] points in the same direction, identifying denitrification as the most likely pathway for $N_2O$ production in larval guts. Whereas our data base was too inconsistent for upscaling, Majeed et al. [55] attributed 0.2–1.8% of $N_2O$ emissions from tropical soils to Scarabaeidae larvae. In contrast to our study, they observed consistent $N_2O$ emissions from their laboratory-incubated larvae with an average emission rate of 10 ng $N_2O$ $g^{-1}$ larval fresh weight $h^{-1}$.

## Conclusions

Overall, our data show that Scarabaeidae larvae should not be neglected as sources of $CO_2$, $CH_4$, and $N_2O$ in soil GHG flux research. However, to assess the impact of Scarabaeidae larvae on regional and global GHG budgets and to better understand seasonal and interannual variations in GHG release, including the possibility of increased $CH_4$ consumption in soils, it is

mandatory to gather more field data on (gross) production rates and species-dependent spatial larval biomass distributions and activities. These are exactly the same challenges that are known from termite GHG emission research [35], but we have to address these challenges if we want to explicitly include these soil faunal groups in future ecosystem and Earth system models.

## Supporting information

**S1 Fig. Analysis of larval field emissions (on the level of the individual larvae).** A test for correlations between paired samples was performed on the entire pooled larval field dataset using Spearman's *rho* statistic. Shown are the correlations between $CO_2$ emission (mg $h^{-1}$ $larva^{-1}$), $CH_4$ emission ($\mu g\ h^{-1}\ larva^{-1}$), larval biomass (g), larval excavation depth (cm below soil surface), air temperature (˚C), and incubation time (in minutes).
(PDF)

**S1 File. *Melolontha* spp. greenhouse gas emission data.** The complete dataset presented in this study. Sampling site specifications, larval characteristics and greenhouse gas emission values for each single larva. Furthermore, the complete flux calculation procedure has been included for each gas species.
(ODS)

**S2 File. Regression analysis of cumulated larval $CO_2$ and $CH_4$ emissions estimates (plot level).** This file shows the results of the regression analysis which are shown in abbreviated form in Fig 2 in the paper. Cumulated larval $CO_2$ and $CH_4$ emission estimated from field measurements for each individual sampling plot were regressed on the total cumulated plot larval biomass.
(PDF)

## Acknowledgments

We would like to thank Frauke Dormann (Geisenheim University) for the gas chromatographic analysis of the gas samples. We would also like to thank the following persons for access to the collection sites, help during the excavation of larvae, and ecological insights into these animals: Annemarie Peters (Hochschule Geisenheim University), Rainer Hurling and Sabine Weldner (NW-FVA–Northwest German Forest Research Institute), Wolfgang Dieminger and Hermann Stolz (local administration Blaubeuren-Weiler), Jana Reetz and Matthias Inthachot (LTZ–Center for Agricultural Technology Augustenberg), Anne-Katrin Möller (district administration Alb-Donau-Kreis), Gregor Seitz (FVA–Forest Research Institute Baden-Württemberg) and Norbert Kelm (forester municipality Iffezheim).

## Author Contributions

**Conceptualization:** Carolyn-Monika Görres, Claudia Kammann.

**Formal analysis:** Carolyn-Monika Görres.

**Funding acquisition:** Carolyn-Monika Görres, Claudia Kammann.

**Investigation:** Carolyn-Monika Görres.

**Project administration:** Carolyn-Monika Görres.

**Resources:** Claudia Kammann.

**Supervision:** Claudia Kammann.

**Writing – original draft:** Carolyn-Monika Görres.

**Writing – review & editing:** Carolyn-Monika Görres, Claudia Kammann.

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
