## [Decision Letter · Decision Letter 0]

19 Mar 2020

PONE-D-20-01452

First field estimation of greenhouse gas emissions from European soil-dwelling Scarabaeidae larvae targeting the genus *Melolontha*

PLOS ONE

Dear Dr. Görres,

Thank you for submitting your manuscript to PLOS ONE. After careful consideration, we feel that it has merit but does not fully meet PLOS ONE’s publication criteria as it currently stands. Therefore, we invite you to submit a revised version of the manuscript that addresses the points raised during the review process.

We would appreciate receiving your revised manuscript by May 03 2020 11:59PM. To enhance the reproducibility of your results, we recommend that if applicable you deposit your laboratory protocols in protocols.io, where a protocol can be assigned its own identifier (DOI) such that it can be cited independently in the future. For instructions see: http://journals.plos.org/plosone/s/submission-guidelines#loc-laboratory-protocols

We look forward to receiving your revised manuscript.

Kind regards,

Debjani Sihi

Academic Editor

PLOS ONE

Journal Requirements:

Reviewers' comments:

Reviewer's Responses to Questions

**Comments to the Author**

1. Is the manuscript technically sound, and do the data support the conclusions?

Reviewer #1: Partly

Reviewer #2: Partly

2. Has the statistical analysis been performed appropriately and rigorously? 

Reviewer #1: No

Reviewer #2: Yes

3. Have the authors made all data underlying the findings in their manuscript fully available?

Reviewer #1: Yes

Reviewer #2: No

4. Is the manuscript presented in an intelligible fashion and written in standard English?

Reviewer #1: Yes

Reviewer #2: Yes

5. Review Comments to the Author

Reviewer #1: Dear Dr. Görres and Dr. Kammann,

My name is Angel S. Fernandez-Bou (I prefer to disclose my name to improve transparency of the review process). I have worked for a few years with CO2 and greenhouse gas emissions from leaf-cutter ant nests in Central America. Our work has followed an environmental engineering approach to measure and model the soil CO2 dynamics (concentrations and fluxes, and more modestly other greenhouse gases) and their scalability.

I have attached my review as a PDF file for your convenience, although I am pasting it here as well. I hope it is useful for you.

Best luck with your research.

Angel S.

Review of “First field estimation of greenhouse gas emissions from European soil-dwelling Scarabaeidae larvae targeting the genus Melolontha”

In this study, the authors incubated beetle larvae in sealed glass tubes to measure their greenhouse gas emissions (mainly CO2 and CH4). The focus were the measurements occurred in the field, but the authors also conducted laboratory measurements for comparison purposes. Then, they upscaled the results to provide a more holistic understanding of the impact of those emissions. The main finding (in my opinion) is that laboratory incubations underperformed compared to field measurements, stressing the relevance of conducting field work.

This manuscript is well written in general, and mostly easy to follow for a general educated audience. However, I have a few questions and concerns about the methodology that I hope the authors can clarify. My main concern is that these measurements must be extremely precise, since the GHG emissions of a larva in about one hour are very small. Then, a very minor inconsistency may lead to a major error when upscaled. I also disagree with the wording “CH4 emissions”, since (as the authors acknowledge) that methane may be consumed in the soil. Therefore, “CH4 production” sounds more accurate in my opinion.

I have listed (1) the major questions that may require just clarification, but they are key for me to be able to recommend the acceptance of the manuscript; (2) several minor comments and suggestions that are up to the authors to integrate (or not) in their manuscript; and (3) some grammatical and standardization comments that I hope can help.

Major questions

Lines 69 and 70. There are several field studies about greenhouse gas emissions from ants in different regions. Certainly not enough, but there are some good examples:

- Roces, Bollacci, Forti, and Kleineidam have done several field studies measuring CO2 emissions from massive nests in Brazil and South America, even creating casts of the nests.

- Fernandez-Bou, Harmon, and Dierick have two field studies about CO2 emissions and soil CO2 dynamics related to leaf-cutter ants that were upscaled to the ecosystem scale in Central America.

- Soper found remarkable N2O emissions from refuse piles of Atta colombica in Costa Rica;

- Emissions from smaller ant nest in northern Europe have been studied by Finér, Risch, Ohashi; Ohashi also has at least a publication with emissions from ants in Thailand.

Lines 123 to 130. Were the vacutainers filled at the same moment than the larvae were placed in the containers for incubation? In other words, were the vacutainers filled with air containing the exact same GHG concentrations that what the larvae containers had in the beginning of the incubation period? This is how the authors calculate the GHG production by the larvae, and if the initial concentrations were not the same, the production would not be reliable. If this is the case, I suggest they rephrase it to something like “…ambient air at the beginning of the field incubations.”. This is a major concern for me, since I do not understand why some N2O flux measurements are negative.

Ln 124 to 125. At what depth were the larvae incubated? Were they incubated at the depth they were found? And did the authors measure the temperature at that depth? It occurs to me that, since temperature at different depths is usually different, changing the temperature at which the larvae were found may have changed their regular metabolism.

Lines 167 to 175. If I understood correctly, you used correlation between pairs of variables. Why? I would probably have used a generalized linear mixed model (glmmPQL, MASS package in R), since some variables can influence and be influenced by others, and interactions among them should be considered. In addition, the plot seems to be a random factor, since there are characteristics intrinsic of the plot for which the measured variables do not account (I like the example of the researcher who asks questions to people; fixed factors are sex, age, education, etc., while the random factor is the person identification, since they have their own personal background that cannot be measured by the fixed factors but influences their response to some extent). I do not mean that GLMM are the only way to do it, but I would like to understand a solid reason why you chose to conduct the correlation analysis.

Also, if you are not familiar with GLMM and are curious, there is a nice manual by Bodo Winter about LMM (https://arxiv.org/ftp/arxiv/papers/1308/1308.5499.pdf).

Lines 178 to 189. Similarly to the comment before, why did you do a simple linear regression considering only one independent variable?

Upscaling. I would have measured the CO2 and CH4 flux in the field before excavating the plots. This is a major part of the methodology that I think is missing here, in particular to solidify the possible CH4 emission claims. As the authors point out, increased CH4 production leads to the possibility of increased soil CH4 consumption. And I think the authors agree with this in the conclusions, when they say, “Scarabaeidae larvae should not be neglected as sources of CO2, CH4, and N2O in soil GHG flux research.”. They did not measure soil GHG flux in this study and that would have made a more solid study in my opinion, but I do not think that is a reason to reject this manuscript.

The authors discuss and acknowledge the importance of “the effect” of CH4 production by soil fauna instead of the faunal CH4 emissions. And to be consistent, in the upscaling results, I think it is more correct to speak about CH4 production and not about CH4 emissions, since that CH4 may not reach the atmosphere as it may be consumed before (therefore, not emitted). This may affect the emissions claims presented in the study and even the title.

Lines 322 and 333. I disagree. The nature of CH4 emissions by termites is completely different than the potential (if any) soil CH4 emissions by beetles. As stated before and by the authors, CH4 produced in the soil matrix is normally consumed as it diffuses through a tortuous and long path, while CH4 produced in the guts of the termites and by their nest metabolism is connected with the atmosphere by the cathedral vents and transported by convection, a mechanism several orders of magnitude more efficient that soil GHG diffusion. Then, CH4 produced by termites finds (quite fast) its way to the atmosphere, while that produced by Scarabaeidae likely not.

Minor comments, suggestions, and questions

Line 96. I think it would be useful to mention regular size and weight of the larvae, for those readers not familiar with them.

Lines 109 to 120. Did the authors collect all the larvae present in the plots to account for abundance/density? Is this how they were able to upscale? If yes, I suggest clarifying it in the “Soil excavations” section.

Line 179. What is (or how are the calculations for) the larval density per unit area?

Line 184. What is the literature? Can you add citations and mention the numbers here?

Lines 195 and 196. Here I miss the interaction among variables. For example, does the temperature influence larval biomass, hence indirectly CO2 production?

Line 200. I disagree with using “CH4 emissions”; larval “CH4 production” is more accurate.

Line 223. Larval abundance methods are not clearly explained in the methodology.

Line 232. References?

Line 234. What is the CO2 efflux por unit area? And CH4 results should refer to production and not emissions.

Line 234. "in this region" is confusing, since the authors are referring to the European level. Where in Europe (approximately) are they talking?

Line 245 to 246. That is interesting! Also good point in lines 268 and 269.

Line 250 to 252. Why do the authors think only some larvae produce N2O? Can they add that in the discussion?

Lines 274 to 282. I think this should be revised in line with all the suggestions and comments.

Lines 289 to 307. Very good discussion.

Lines 308 to 314 and S1_File. Why the authors think N2O flux is negative in a few occasions?

Line 310. Not “considerable,” but for ants there is some literature about N2O (Soper et al, 2019, etc.).

S1_File. The authors do not report the time of sampling, but I wonder if they would have found a diel pattern, for example, consistent with the soil CO2 diel pattern: when CO2 concentration in the soil is higher, do the larvae produce the same or less, since there is less O2?

Grammar and standardization

I kindly suggest that the authors revise the manuscript for consistency in at least three points that I noticed:

1. The authors did not use the serial comma in some occasions, but they did in others. I suggest homogenizing the whole document using the serial comma. For example, in lines 22, 57, 58, 60, etc.

2. A comma should not separate the subject and the verb. For example, in line 20, I believe that the correct way to write this is "…from laboratory experiments is often controversial, and it has been limited...". In lines 48 and 49 “… (N2O), but they are also…”. Other examples of missing subject after comma are in lines 64, 65, 133, etc.

3. I suggest following the International System units and rules. A checklist can be found online (I use this one https://physics.nist.gov/cuu/Units/checklist.html). For example, in lines 31 and 32, units must be provided after the amounts, and I would use “Gg” (giga grams) instead of “kt” (unless the authors think those units are better for broader communication): “10.42 Gg CO2 yr-1 and 409.53 Gg CO2 yr 1, and 0.01 Gg CH4 yr-1 and 1.36 Gg CH4 yr-1.”. The same occurs in line 74, 88, 89, 99, 100, 149, 150, 163, etc. In line 104, there should be a space character in “1 m”. In line 109, I believe the authors want to use “×” (Alt + 0215) instead of “x”. In lines 127, 128, and 132, change “ml” to “mL”. Line 146, use “Pa” or “kPa” instead of “PSI”. And there may be other excerpts needing standardization that the authors will likely find quickly with the help of the checklist.

References

Bollazzi M, Forti LC, Roces F (2012) Ventilation of the giant nests of Atta leaf-cutting ants: does underground circulating air enter the fungus chambers? Insect. Soc 59:487–498. https://doi.org/10.1007/s00040-012-0243-9

Kleineidam C, Roces F (2000) Carbon dioxide concentrations and nest ventilation in nests of the leaf-cutting ant Atta vollenweideri. Insect Soc 47:241–248. https://doi.org/10.1007/PL00001710

Kleineidam C, Ernst R, Roces F (2001) Wind-induced ventilation of the giant nests of the leaf-cutting ant Atta vollenweideri. Naturwis-senschaften 88:301–305. https://doi.org/10.1007/s001140100235

Fernandez‐Bou, A. S., Dierick, D., Swanson, A. C., Allen, M. F., Alvarado, A. G. F., Artavia‐León, A., et al. (2019). The role of the ecosystem engineer, the leaf‐cutter ant Atta cephalotes, on soil CO2 dynamics in a wet tropical rainforest. Journal of Geophysical Research: Biogeosciences, 124, 260– 273. https://doi.org/10.1029/2018JG004723

Fernandez-Bou, A.S., Dierick, D. & Harmon, T.C. Oecologia (2020). Diel pattern driven by free convection controls leaf‑cutter ant nest ventilation and greenhouse gas emissions in a Neotropical rain forest. https://doi.org/10.1007/s00442-020-04602-2

Soper FM, Sullivan BW, Osborne BB et al (2019) Leaf-cutter ants engineer large nitrous oxide hot spots in tropical forests. Proc R Soc B Biol Sci 286:20182504. https://doi.org/10.1098/rspb.2018.2504

Ohashi M, Finér L, Domisch T et al (2005) CO2 efflux from a red wood ant mound in a boreal forest. Agric For Meteorol 130:131–136. https://doi.org/10.1016/j.agrformet.2005.03.002

Risch A, Schuetz M, Jurgensen MF et al (2005) CO2 emissions from red wood ant (Formica rufa group) mounds: seasonal and diurnal patterns related to air temperature. Ann Zool Fenn 42:283–290. https://www.jstor.org/stable/23735916

Reviewer #2: This is an interesting topic, how Scarabaeidae larvae can affect GHG emissions from soils. These emissions by the soil animals are not well studied, and therefore this manuscript would increase our knowledge about the topic. However, I am concerned about few points.

1. It is not clear how the emissons from the larvae were actually calculated. One gas sample was taken about 1 hour (why not exactly the same time always?) after sealing the animal in a bottle and then ambient sample was taken (not described how) and then there was a blank? Empty bottle? Was the emisson calculated just by dividing the final concentration by time? (Measured concentrations are not given in the manuscript). If so this might give the wrong result or not totally comparable results if the closing time is different.

2. When the larvae were taken out from their natural environment they might be stressed and it would affect perhaps their behaviour. This could explain differences between measurements in the field and in the laboratory. BUT I would not call the measurements in the field "Field measurements" because the animals are taken out from their natural habitat.

3. Based on 1 and 2 I would be very careful in upscaling the results, also the small number of sites and one sampling time and lack of real field measurements are affecting the quality of the data.

Some minor comments:

-were the larvae cleaned /washed from soil before measurements...? Or measured with soil?

-Line 134 Why carrot was given, do they eat carrot in nature?

-Line 135 tap water (not tab)

-Line 149-150 calibration was done with very high concentrations, you did not show the concentrations measured from the bottles, were they in the range of standards?

-Is Fig 2 repeating the same data as in Fig 1?

-Supplement Figure1, This figure is quite complicated to read, It does not open to me, could you split the data into several tables/figures. Units are missing from the scales!

6. PLOS authors have the option to publish the peer review history of their article (what does this mean?). If published, this will include your full peer review and any attached files.

Reviewer #1: Yes: Angel S. Fernandez-Bou

Reviewer #2: No

---

## [Author Response · Author response to Decision Letter 0]

11 May 2020

Please find all responses in the supplied file "Response to Reviewers".

---

## [Decision Letter · Decision Letter 1]

22 Jun 2020

PONE-D-20-01452R1

First field estimation of greenhouse gas release from European soil-dwelling Scarabaeidae larvae targeting the genus *Melolontha*

PLOS ONE

Dear Dr. Görres,

Thank you for submitting your manuscript to PLOS ONE. After careful consideration, we feel that it has merit but does not fully meet PLOS ONE’s publication criteria as it currently stands. Therefore, we invite you to submit a revised version of the manuscript that addresses the points raised during the review process.

We look forward to receiving your revised manuscript.

Kind regards,

Debjani Sihi

Academic Editor

PLOS ONE

Reviewers' comments:

Reviewer's Responses to Questions

**Comments to the Author**

1. If the authors have adequately addressed your comments raised in a previous round of review and you feel that this manuscript is now acceptable for publication, you may indicate that here to bypass the “Comments to the Author” section, enter your conflict of interest statement in the “Confidential to Editor” section, and submit your "Accept" recommendation.

Reviewer #1: All comments have been addressed

Reviewer #3: All comments have been addressed

2. Is the manuscript technically sound, and do the data support the conclusions?

Reviewer #1: Yes

Reviewer #3: Yes

3. Has the statistical analysis been performed appropriately and rigorously? 

Reviewer #1: Yes

Reviewer #3: Yes

4. Have the authors made all data underlying the findings in their manuscript fully available?

Reviewer #1: Yes

Reviewer #3: (No Response)

5. Is the manuscript presented in an intelligible fashion and written in standard English?

Reviewer #1: Yes

Reviewer #3: Yes

6. Review Comments to the Author

Reviewer #1: Hello again, Dr. Görres and Dr. Kammann. I hope you are doing well.

I think you have done a very nice job responding to the reviewers' questions and addressing the comments. I also appreciate your clarifications across the manuscript thinking about a broader audience.

In my opinion, your manuscript is ready for publication.

Best luck on your future research.

Angel S.

Reviewer #3: See below:

Comments to the Authors First field estimation of greenhouse gas emissions release from 2 European soil-dwelling Scarabaeidae larvae targeting the genus Melolontha The manuscript is well written and has concise, well connected sentences. In this study, the authors have made an attempt to do field and lab scale measurement of greenhouse gas emissions from Scarabaeidae larvae. I really liked the idea of upscaling the results in order to provide better understanding of the impacts of such emissions. Several studies talk about microbial decomposition of soil organic carbon and how that influences the atmospheric greenhouse gas concentrations but very few have talked about other faunal groups such as ants, termites and beetle larva. So, this is a good attempt to quantify that. I have gone through the responses to reviewers’ comments and most of them seem satisfactory to me. Some of my overall concerns on the methodology of this manuscript are below. The larvae were incubated at the soil surface in field, but the larvae were actually excavated from deeper depths. Do you think by doing so, we are adding a bias to the greenhouse gas emissions that actually comes from larvae at deeper depths versus what is reported in the manuscript? How the calculations for of these emissions were done? Also, the larvae used in field study and lab incubation, were they all similar in size? If not, do you think that would influence the greenhouse gas emissions as well? Other than these minor concerns, I have gone through the responses to the reviewers and they seem convincing and satisfactory to me. 

7. PLOS authors have the option to publish the peer review history of their article (what does this mean?). If published, this will include your full peer review and any attached files.

Reviewer #1: Yes: Angel Santiago Fernandez Bou

Reviewer #3: No

---

## [Author Response · Author response to Decision Letter 1]

31 Jul 2020

See attached file "Response to Reviewers.docx".

---

## [Editor Report · Decision Letter 2]

10 Aug 2020

First field estimation of greenhouse gas release from European soil-dwelling Scarabaeidae larvae targeting the genus Melolontha

PONE-D-20-01452R2

Dear Dr. Görres,

We’re pleased to inform you that your manuscript has been judged scientifically suitable for publication and will be formally accepted for publication once it meets all outstanding technical requirements.

Kind regards,

Debjani Sihi

Academic Editor

PLOS ONE
---

## [Editor Report · Acceptance letter]

17 Aug 2020

PONE-D-20-01452R2 

First field estimation of greenhouse gas release from European soil-dwelling Scarabaeidae larvae targeting the genus *Melolontha*

Dear Dr. Görres:

I'm pleased to inform you that your manuscript has been deemed suitable for publication in PLOS ONE. Congratulations! Your manuscript is now with our production department. 

Kind regards, 

on behalf of

Dr. Debjani Sihi 

Academic Editor

PLOS ONE